# Improvement of Phosphate Adsorption Kinetics onto Ferric Hydroxide by Size Reduction

**Vicenç Martí** [1,2,*]**, Irene Jubany** [2]**, David Ribas** [2]**, José Antonio Benito** [3] **and Berta Ferrer** [1]

1 Barcelona Research Center in Multiscale Science and Engineering (EEBE),
Department of Chemical Engineering, Technical University of Catalonia (UPC),
Av. Eduard Maristany 16, 08019 Barcelona, Spain; bferrerorra@gmail.com
2 Eurecat, Centre Tecnològic de Catalunya, Sustainability Area, Plaça de la Ciència 2, 08243 Manresa, Spain;
irene.jubany@eurecat.org (I.J.), davidrrff@gmail.com (D.R.)
3 Department of Materials Science and Metallurgical (EEBE), Technical University of Catalonia (UPC),
Av. Eduard Maristany 16, 08019 Barcelona, Spain; jose.a.benito@upc.edu
* Correspondence: vicens.marti@upc.edu; Tel.: +34-93-401-0957

**Abstract:** Ball milling and ultra-sonication size reduction procedures were applied to granular ferric hydroxide (GFH) to obtain two micro-sized adsorbents. These two adsorbents and GFH were investigated to improve the removal of phosphates from water. The size reduction procedures, using the milling method, allowed a reduction of size from 0.5–2 mm to 0.1–2 $\mu$m and total disaggregation of the GFH structure. Using an ultra-sonication method yielded a final size of 1.9–50.3 $\mu$m with partial disaggregation. The Langmuir model correlated well with the isotherms obtained in batch equilibrium tests for the three adsorbents. The maximum adsorption capacity ($q_{max}$) for the milled adsorbent was lower than GFH, but using ultra-sonication was not different from GFH. The equilibrium adsorption of two wastewater samples with phosphate and other anions onto the GFH corresponded well with the expected removal, showing that potential interferences in the isotherms were not important. Batch kinetics tests indicated that the pseudo second-order model fitted the data. Long-term adsorption capacity in kinetics ($q_e$) showed the same trend described for $q_{max}$. The application of milling and ultra-sonication methods showed 3.5- and 5.6-fold increases of the kinetic constant ($k_2$) versus the GFH value, respectively. These results showed that ultra-sonication is a very good procedure to increase the adsorption rate of phosphate, maintaining $q_e$ and increasing $k_2$.

**Keywords:** adsorption technology; ultra-sonication; phosphate removal; granular ferric hydroxide; micro-sized adsorbents

## 1. Introduction

Several Circular Economy approaches applied to the adsorption of phosphate have been considered in the literature. These alternatives include the recovery of phosphorus and water from several wastewater streams [1–4], the use of low-cost products or recovered waste as sorbents [5–11], and the study of the re-use of phosphate-loaded sorbents [2,3,11–13]. To develop these strategies, it is paramount to optimize the performance of the sorption process.

Commercial granular ferric hydroxide (GFH) is formed by porous solids that have demonstrated a very good sorbent capacity for the recovery of phosphates from water in batch systems [1,12,14–17]. The adsorption of phosphates from aqueous solutions and wastewaters onto the GFH is usually performed in fixed-bed type contactors [18]. This system is easily operated, but multiple references show early breakthrough curves of phosphate in fixed-bed column assays [14,19,20]. This behavior is linked with sorbent capacities lower than in batch tests and is attributed to kinetic mass-transfer limitations [11,14,21]. Several ferric hydroxide sizes below the usual GFH size range (1 mm) have been used as promising materials to increase the kinetics and adsorption capacity of phosphate

in stirred batch tests, as it happens, for instance, in the case of activated carbon with micro-contaminants [22,23]. The procedures involved for size reduction of the adsorbent include sieving of GFH [24], grinding [1,17], or the synthesis and use of nano-sized materials [19,24–26]. From these references, only one study was centered in the role of reduced size particles obtained from the same bulk GFH on the adsorption of phosphate (equilibrium and kinetics) [17]. This specific study showed that all adsorbents exhibit similar adsorption capacities at equilibrium and that a reduction of sizes increased the kinetic constant of the initial sorbents up to 100-fold.

Top-down methods such as high energy ball milling are very effective alternatives that have been used for size reduction other water treatment materials such as nano-zero valent iron [27,28] and calcite [29]. Ultra-sonication is another top-down size reduction method able to produce microparticles and nanoparticles directly from some soft bulk material as hematitic/goethitic iron ore fines [30], talc [31], or $TiO_2$ [32]. Milling and ultra-sonication methods have the advantage that the chemistry of the particles will not be altered, hence bulk and particle adsorbents could be more directly comparable.

Within this context, the aim of this work is two-fold: (i) size reduction of one low-cost commercial GFH, using ball-milling and ultra-sonication, as a new method, to obtain low-size adsorbents; and (ii) the comparison of these materials of different sizes in terms of phosphate adsorption capacity and kinetic behaviour at the laboratory scale. Within this context, this study aims to understand the effects of size reduction on the improvement of sorption capacity and/or kinetics.

## 2. Materials and Methods

### 2.1. Adsorbent Preparation

The commercial GFH adsorbent used was Ferrosorp GW® (HeGo Biotec GmbH, Giessen, Germany) with sizes in the range of 0.5–2 mm. This kind of adsorbent mainly consists of $Fe(OH)_3$ and is obtained from an industrial low-cost product [12].

Two methods were used for the size reduction of the GFH solid.

The first method involved a two-step milling process, in which the first step consisted of dry milling, using a hammer mill (MF 10 basic microfine grinder drive, IKA-Werke GmbH, Staufen Germany) at 6000 rpm. Then, 250 g of the GFH sample was passed through this mill three times and sieved at 100 μm. In the second milling step, 6 g of this pre-milled sample was placed in a 250 mL carbon steel vial, alongside 200 g of S110 spherical high-carbon steel shots (Pometon S.p. A, Italy) and 150 mL of DI water. The material was milled at 250 rpm for 5 h using a Planetary ball mill (Fritsch GmbH, P-5, Mahlen und Messen, Germany). After milling, the slurry was separated from the steel grinding balls using a 75 μm sieve. The retained material was washed several times with DI water over the recovered slurry until reaching a final volume of 500 mL. The slurry was dried overnight in an oven at 100 °C (J.P. Selecta, Digiheat, Abrera, Spain) and the dry solid was disaggregated using an agate mortar. This milled solid is referred to as OF-M in the present work.

The second method used for the size reduction was disaggregation by ultrasonic waves. For this purpose, around 5 g of solid GFH was mixed with DI water in a 250–500 mL volumetric flask and sonicated for 5 min at 20 kHz in a lab ultrasonic cleaner (ATU Ultrasonidos, ATM40-2L-CD, Paterna, Spain). The suspension was centrifuged at 4000 rpm for 5 min to remove supernatant water. The solid was dried and disaggregated like in the milling method (see above). The dry particles obtained from this ultrasonic-based protocol are referred to as OF-U.

### 2.2. Characterization of Adsorbents

The content of total iron and calcium in the GFH sample was determined by sieving the dried solid below 40 μm and sampling 50 mg for acid digestion using HCl 30% Suprapure® (Supelco, Germany). After filtering the digested solid with a 0.45 μm filter, the total iron and calcium contents were obtained using an AAS instrument (Analytik Jena GmbH, contrAA 800, Jena, Germany).

The general morphology and individual particle size were studied by scanning electron microscopy (SEM) (Zeiss, Gemini ultra plus, Jena, Germany) equipped with energy-dispersive X-ray spectroscopy (EDS) (Oxford Instruments, X-MAX 50 mm$^2$, Abingdon, UK). Sample preparation involved two steps: (I) deposition and room evaporation over aluminium pins, and (II) sample metallization with Au-Pd 30 s at 18 mA (Quorum, Emitech SC7620 mini sputter, Laughton, UK).

The suspensions' granulometry was analysed using laser diffraction spectrometry for the OF-M samples (Beckman Coulter, LS 13320, Brea, CA, USA) and with a Malvern Panalytical Mastersizer 3000 (Malvern Panalytical Ltd., UK) for the OF-U samples in DI water.

The nitrogen Brunauer–Emmett–Teller (BET) specific surface area (SSA) was measured (Micromeritics ASAP 2020, Aachen, Germany). Degassing was carried out for several hours at a maximum temperature of 100 °C. Sample preparation for this analysis involved drying 1 g of the slurry. It was performed at 60 °C in a biological incubator (Raypa, Incuterm Digit, Terrassa, Spain) for 72 h. Thereafter, a dried cake was obtained, which was crushed to a fine powder using a manual agate mortar. The above-mentioned drying procedure was performed in a negative pressure lab cabinet to avoid exposure to nanopowder.

The zeta potential of 1 g/L suspensions of OF-M and OF-U was measured using 0.01 M analytical grade NaCl (Panreac, Spain) at equilibrium pH and dynamic light scattering detection (Brookhaven Instruments, NanoBrook Zeta Pals; Holtsville, NY, USA).

The pH of the point of zero charge (PZC) of GFH was evaluated via the immersion technique [33] using 25 g/L suspensions with an initial pH adjusted in the range of 3–12 with analytical grade hydrochloric acid (HCl) or sodium hydroxide (NaOH) (Panreac, Spain), as measured with a pH-meter (Crison, GLP 22, Alella, Spain).

### 2.3. Batch Adsorption Procedure

Phosphate test solutions were prepared by weighing $K_2HPO_4$ analytical grade (Scharlau, Spain). The dissolutions and dilutions were performed using deionized (DI) water (Merck-Millipore, Elix 70, Darmstadt, Germany). The pH was adjusted to 8.0 using HCl. This initial value of pH was chosen as a compromise between the buffer capacity of phosphate and the observed equilibrium pH. In this way, the theoretical predominant speciation of phosphate in all the synthetic samples was $HPO_4^{2-}$. Phosphate equilibrium adsorption with GFH was also tested in two types of treated wastewater. The first wastewater (MR-1) was generated during the cleaning operation with phosphate products in a pig slaughterhouse and the treatment of water applied consisted of coarse and fine screening, homogenization, flotation, and nitrification-denitrification biological treatment. The second wastewater (MR-2) was obtained from a truck cover manufacturing textile company in which phosphate was associated with manufacturing process and cleaning operations. Treatment for MR-2 included physicochemical treatment, secondary biological treatment, and tertiary treatment with activated carbon.

Equilibrium and kinetic batch tests were performed by placing 50 mL of the phosphate dissolutions in Falcon tubes with different weighted amounts (Sartorius AG, Practum 513-S, Göttingen, Germany) of each adsorbent at room temperature. Then, stirring at 8 rpm (Heidolph GmbH, REAX 20, Schwabach, Germany) was performed under dark conditions. After stirring the mixtures, the suspensions were centrifuged (J.P. Selecta, Centronic-BLT, Abrera, Spain) at 4000 rpm for 5 min. The supernatant solution was filtered with a regenerated cellulose (RC) 0.2 μm filter and analyzed using ion chromatography (Dionex, ICS 2100, Sunny Valley, OR, USA) with a 4 × 250 mm Ion Pac AS19 Column. The analysis method used 1 mL/minute of the generated KOH mobile phase with a gradient of 10–35 mM. The quantitation limit of the method was 0.5 mg $PO_4^{3-}$/L. All masses of phosphate in the present work are expressed as mg $PO_4^{3-}$.

The adsorption of phosphate onto the materials was calculated from the following mass balance equation:

$$q = \frac{(C_o - C) \cdot V}{m} \tag{1}$$

where q is the amount of phosphate adsorbed onto the solid at time t (mg/g); $C_o$ and C are the phosphate concentrations at time zero and at time t (mg/L), respectively; V is the volume of dissolution in the batch experiment (L); and m is the mass of the solid adsorbent in the experiment (g).

The removal of phosphate, η (mass %), is given in Equation (2) and can also be expressed in terms of batch experiment conditions using Equation (1):

$$\eta = \frac{100 \cdot (C_o - C)}{C_o} = \frac{100 \cdot m \cdot q}{V \cdot C_o} \tag{2}$$

Equilibrium batch tests were performed using replicate experiments of phosphate dissolutions (10, 20, 33, 50, 90, and 150 mg/L) with different weighed amounts of the three adsorbents (1, 2, and 3 g/L) for 120 h. The wastewater samples were filtered with a 0.45 μm nylon filter before the equilibrium experiment, which was performed with 2 and 3 g/L of GFH for 120 h and analyzed by ion chromatography, as mentioned before. Chloride, nitrate, and sulfate concentrations were also determined to check potential competition of these frequent anions with phosphates.

Freundlich (Equation (3)) and Langmuir (Equation (4)) isotherms were used to fit experimental equilibrium data $C_e$ and $q_e$ (obtained from the corresponding equilibrium using Equation (1)). These models have been originally applied in the literature [34,35].

$$q_e = K_F \cdot C_e^{1/n} \tag{3}$$

$$q_e = q_{max} \cdot \frac{C_e \cdot b}{1 + b \cdot C_e} \tag{4}$$

where $K_F$ $(mg^{1-1/n} \cdot (L)^{1/n}/g)$ and n are the constants of the Freundlich model that can be obtained by fitting log $C_e$ versusvs log $q_e$. In the Langmuir isotherm, $q_{max}$ is the maximum adsorption capacity (mg/g) and b is the binding constant (L/mg). Both parameters can be obtained by rearranging Equation (4):

$$\frac{C_e}{q_e} = \frac{1}{q_{max} \cdot b} + \frac{C_e}{q_{max}} \tag{5}$$

The theoretical calculation of phosphate removal was performed from the equilibrium concentration of GFH isotherm using Equations (1), (2),) and (4), coded into a Microsoft Excel™ spreadsheet.

The kinetic behaviour of the three adsorbents was determined through duplicate kinetic batch tests with 2 g/L of the materials and 30 mg/L phosphate, whichthat is the minimum expected concentration in the wastewater to be analyzed. Test tubes were extracted after 1, 2, 3, 4, 5, 6, 22, 24, 27, 48, 52, 72, 75, 96, 144, and 168 h of contact time.

Pseudo first- (Equation (6)) and pseudo second-order (Equation (7)) models were employed to identify batch kinetics:

$$q = q_e \cdot (1 - \exp(-k_1 \cdot t)) \tag{6}$$

$$q = \frac{k_2 \cdot q_e^2 \cdot t}{1 + k_2 \cdot q_e \cdot t} \tag{7}$$

where $k_1$ is the first-order kinetic constant $(h^{-1})$ and $k_2$ is the second-order kinetic constant (g/mgh).

The corresponding linear forms of pseudo first- and pseudo second-order are as follows, respectively:

$$Ln (q_e - q) = Ln (q_e) - k_1 \cdot t \tag{8}$$

$$\frac{t}{q} = \frac{1}{k_2 \cdot q_e^2} + \frac{t}{q_e} \tag{9}$$

Pseudo second-order kinetics model (Equation (7)) could be linearized up to five different linear forms [36]. Regression using the linear form given in Equation (9) is the most used for phosphate kinetic studies [7–10,15]. This linear form could predict $q_e$ very well, but several references [36–38] indicate that this linearization could be inappropriate to obtain a good value of $k_2$ and the initial adsorption rate value ($k_2 \cdot qe^2$). In order to investigate this question, non-linear models using Equation (7) were applied [39]. The methodology used for non-linear regression consisted of the direct fitting of n data pairs ($t_i$, $q_i$) to Equation (7) using the values of $q_e$ and $k_2$ that minimize the sum of squared errors, $SS_{err}$.

$$SS_{err} = \sum_{i=1}^{n} [q_i - q]^2 \tag{10}$$

The Solver function in Microsoft Excel™ software with the GRG Nonlinear solving method was used to minimize $SS_{err}$ as the objective function to obtain the fitted $q_e$ and $k_2$. The $q_e$ and $k_2$ results obtained in the linear model were used as a starting point for iteration.

*2.4. Statistical Analysis*

Two statistical methods were used to compare the fitting slopes parameters (m) used in linear models (Equations (5) and (9)) in order to investigate the effects of size reduction on $q_{max}$ and $q_e$.

The first method was to obtain the 95% confidence interval (CI) for m, $CI_m$, which is given by the following:

$$CI_m = m \pm t_{(0.025, \, n-2)} \cdot s_e \tag{11}$$

where $t_{(0.025, n-2)}$ is the two-tailed Student's *t*-test value for a significance level of $\alpha = 0.05$ and $n - 2$ degrees of freedom (n is the number of fitting points), and $s_e$ the standard error for the slope m. The extremes of this interval are the lower and the upper confidence levels (LCL and UCL, respectively). The comparison of these $CI_m$ allows detecting the absence of overlap between the adsorbents, and thus confirming the difference between the slope fitting parameters (m).

The second method used was the comparison of the slope parameters of two experiments incorporating a dichotomous (dummy) independent variable, D, [40–43]. This D variable allows the combination of two sets of experimental data that correspond to the two categories (e.g., comparison of kinetics from GFH and OF-U adsorbents using Expression (9)), extending the linear model to a multilinear model in the following form:

$$Y_i = b_o + b_1 \cdot X_i + b_2 \cdot D_i + b_3 \cdot D_i \cdot X_i + error \tag{12}$$

where $Y_i$ and $X_i$ are the linearized data values of the two sets of data (e.g., $t_i/q_i$ and $t_i$ for GFH and OF-U kinetics). The variable $D_i$ takes the value of zero for the data points of the set of reference (e.g., GFH adsorbent) and the value of 1 for the other set of data points (e.g., OF-U).

Testing the statistical significance of $b_3$ parameter in this model, it is possible to know if the slopes of the two sets are equal. In the case where the CI of $b_3$ contains the zero, the hypothesis that the slopes of the two sets are equal cannot be rejected.

In both methods, the fitting calculations and statistical values were obtained using the LINEST function in Microsoft Excel™ software.

## 3. Results and Discussion

*3.1. Characterization of Adsorbents*

The total iron and calcium contents measured in dry GFH were 37.1% (weight to weight) and 6.3%, respectively. These values correspond to 71.0% $Fe(OH)_3$ and 15.6% $CaCO_3$, respectively. The sorbents used in the present paper are said to be formed by $Fe(OH)_3$ and calcite [12]. Therefore, these two compounds would explain more than 85% of the composition of the GFH.

SEM microphotographs of GFH show different superficial structures: big boulders in blue, flake crystal clusters in red, and intricate globular aggregates resembling broccoli in

green (Figure 1b). The structures of the granulated material are lost in the milled material with a size of approximately 100 nm (yellow circles in Figure 1c). Parent material structures are still maintained in the case of ultra-sonication, with sizes of a few microns (Figure 1d).

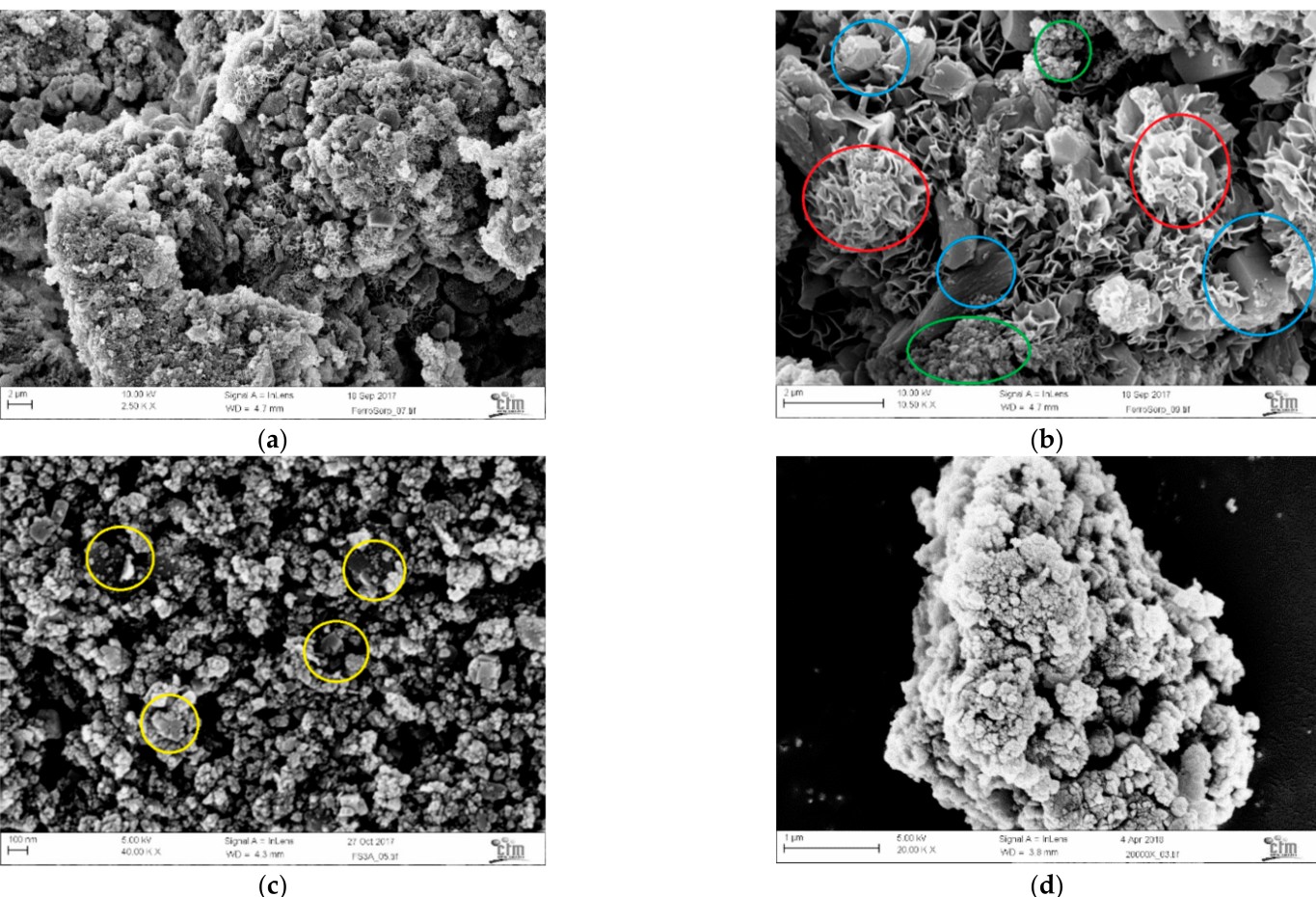

**Figure 1.** Scanning electron microscopy (SEM) microphotographs (**a**,**b**) granular ferric hydroxide (GFH), (**c**) OF-M (milled), and (**d**) OF-U (ultra-sonicated). See main text for explanation of coloured circles.

A summary of the other properties of the materials is shown in Table 1. More detail on the granulometry and BET measurements can be found in the Supplementary Materials (SM).

**Table 1.** Size, Brunauer–Emmett–Teller (BET) area, and zeta potential at equilibrium pH. GFH, granular ferric hydroxide.

|  | **GFH** | **OF-M** | **OF-U** |
|---|---|---|---|
| Size [1] ($\mu$m) | 500–2000 [2] | 0.1–2 | 1.9–50.3 |
| BET Surface Area (m$^2$/g) | 199.7 | 98.4 | 160.2 |
| t-plot Micropore Area (m$^2$/g) | 20.9 | 0 | 4.3 |
| Zeta Potential [3] (mV) | $-8.4 \pm 2.8$ | $-9.5 \pm 1.0$ | $-8.4 \pm 1.9$ |
| Equilibrium pH | 8.59 | 9.41 | 9.43 |

[1] Range p10 to p90 in volume, [2] manufacturer information, [3] average $\pm$ standard deviation.

Size measurements of the OF-M and OF-U in suspension (Figure S1a,b in Supplementary Materials) using DLS (giving the size range between the 10th and 90th percentiles) showed that the milling system worked better for size reduction than the sonication system.

These DLS-based results were confirmed with SEM, although DLS showed possible agglomerations around 2 μm in the OF-M suspensions that were not found using SEM (Figure 1c).

The GFH BET surface area was within the same range as found in the literature (120–300 $m^2/g$) [1,14,15,17,18,44,45]. Comparing the three materials shows the destruction of the BET surface area with size reduction, including an important loss of micropore area, which is 100% in the case of the milled material. The adsorption isotherms of the three materials (Figure S2a–c, in Supplementary Materials) have a similar shape with an important hysteresis and sudden increase of adsorption at high pressures, indicating the presence of a significant amount of mesopores. These results are consistent with the SEM morphology (Figure 1) and indicate that, although milled particles (OF-U) were smaller, the change in surface area is linked to the loss of internal surface and not to an increase in surface area due to size [46].

The zeta potentials at equilibrium pH were very low and similar for all three materials, indicating a slight positive charge of the solid surface, matching the low stability of the suspension (easy sedimentation).

A plot of the change in pH ($\Delta pH$) versus the initial pH for the PZC calculation of the GFH sample shows a very good linear relationship (Supplementary Materials, Figure S3).

$pH_{PZC}$, i.e., the pH that showed no variation of pH ($\Delta pH = 0$), was equal to 8.43. While these values are slightly higher than those reported for akaganeite, which fall within the range of 7.5–8 [14], $pH_{PZC}$ in the GFH sample was very close to equilibrium for zeta potential measurement (see Table 1).

### 3.2. Effect of Particle Size on Adsorption Isotherm

While both the Langmuir and Freundlich models were able to produce acceptable fits to the data from all three adsorbents, the Langmuir model yielded a better overall fit (higher coefficient of determination, $R^2$) (Table 2 and Figure S4a,b in Supplementary Materials).

**Table 2.** Fitting parameters for the Freundlich and Langmuir models.

|  |  | GFH | OF-M | OF-U |
|---|---|---|---|---|
|  |  | Freundlich model | | |
| $R^2$ |  | 0.8847 | 0.9554 | 0.9263 |
| $K_F$ | ($mg^{1-1/n} \cdot (L)^{1/n}/g$) | 11.59 | 9.70 | 12.78 |
| n | (-) | 4.00 | 4.30 | 5.28 |
|  |  | Langmuir model | | |
| $R^2$ |  | 0.9481 | 0.9731 | 0.9358 |
| $q_{max}$ | (mg/g) | 41.80 | 31.59 | 36.43 |
| b | (L/mg) | 0.1006 | 0.1051 | 0.0871 |

Taking the Langmuir model, Figure 2 shows the fitted isotherms for the adsorption data with the three tested adsorbents.

The fit of Figure 2 exhibits a possible loss of adsorption in $q_{max}$ for the reduced materials. Figure S4,b also shows different slopes ($1/q_{max}$). The comparison of $q_{max}$ for the three adsorbents could be performed by the two mentioned statistical analyses and is presented in Table 3.

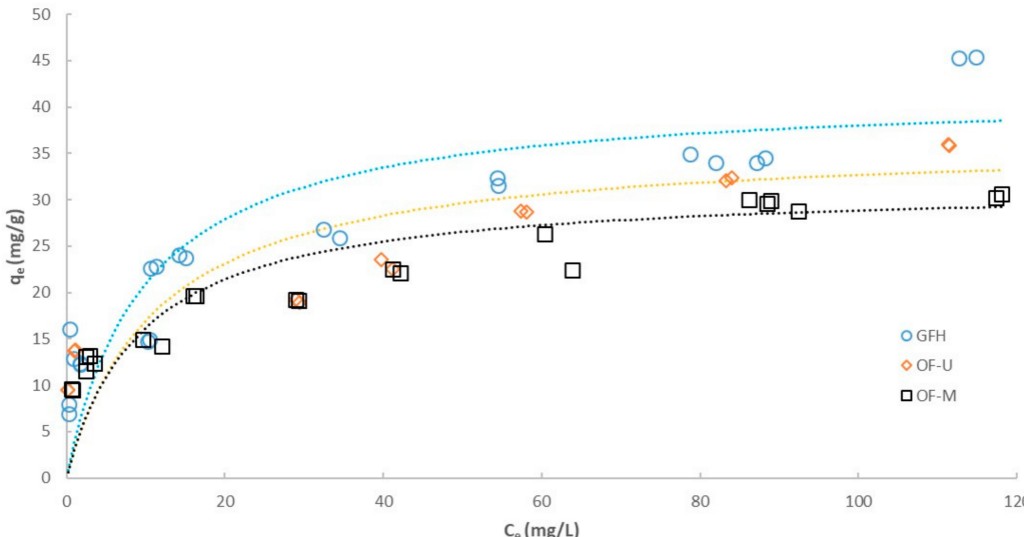

**Figure 2.** Fitted isotherms for the three tested adsorbents using the Langmuir model (dotted lines, see Table 2).

**Table 3.** 95% confidence interval (CI) for $1/q_{max}$ and $b_3$ fitting parameters in the Langmuir model.

|  |  | **GFH** | **OF-M** | **OF-U** |
|---|---|---|---|---|
|  |  | Langmuir model CI | | |
| Slope ($1/q_{max}$) | (g/mg) | 0.023925 | 0.031652 | 0.027449 |
| Standard error | (g/mg) | 0.001251 | 0.001177 | 0.002076 |
| d.f. [1] | - | 20 | 20 | 12 |
| t-Student $_{0.025}$ | - | 2086 | 2086 | 2179 |
| $1/q_{max}$ UCL | (g/mg) | 0.026536 | 0.034107 | 0.031971 |
| $1/q_{max}$ LCL | (g/mg) | 0.021315 | 0.029196 | 0.022926 |
|  |  | Dichotomous model comparison with GFH | | |
| $b_3$ UCL [2] | - | - | 0.011198 | −0.001109 |
| $b_3$ LCL [2] | - | - | 0.004255 | 0.008156 |

[1] degrees of freedom [2] See Supplementary Materials Table S1.

The first method is applied studying the slope of the linear fit, i.e., $1/q_{max}$ and its associated CI (from Equation (10)) for the three adsorbents. Clearly, the resulting $1/q_{max}$ UCL and $1/q_{max}$ LCL differ for OF-M and GFH data because their CIs do not overlap. In the case of OF-U and GFH, there is an overlap of CIs. The second method uses the statistical analysis of the CI for $b_3$ using the dichotomous model (see also Table S1); it could be seen that the comparison of GFH and OF-M slopes showed significant differences, while the comparison of GFH and OF-U slopes resulted in not significant differences in $1/q_{max}$ values. Consequently, the two statistical methods showed that there is a loss of maximum adsorption capacity in the milled solid (OF-M) compared with the GFH adsorbent. This may be partially owing to the loss of specific surface in the OF-M adsorbent because $q_{max}$ in OF-M decreases by around 24% and the loss in surface area is 50%, both with respect to GFH (see Table 1). This loss of adsorption capacity for phosphates using iron hydroxide could be related to the lower specific surface area, but this point is difficult to support for different origins of the adsorbents [24].

In the case of OF-U, $1/q_{max}$ is statistically identical to the value obtained for GFH, despite a 20% loss in surface area. This result is supported by the fact that Ferrosorp adsorbent bellow 100 μm has already shown that the adsorption capacity for phosphates is maintained [17].

The values of $q_{max}$ in Table 2 are better than the typical values for other by-products and wastes such as brick waste [8], copper smelter slag [9], or carbon waste [10] that usually range from 0.04 to 5.35 mg $PO_4^{3-}$/g.

Red mud and commercial akaganeite [15,47], as well as commercial GFH [17,44], exhibit a similar range (26.8 to 39.6 mg $PO_4^{3-}$/g) as shown by the result of this study. Other studies using materials such as impregnated skin split waste [5], hybrid fibres [19], and synthetic materials [26] showed results within the range 66.4 to 220.8 mg $PO_4^{3-}$/g. Akaganeite also showed better results (51.8 to 71.5 mg $PO_4^{3-}$/g) in similar studies [1,24].

### 3.3. Removal of Phosphate from Wastewater

The initial phosphate concentration and GFH dosage for the two types of wastewater (MR-1 and MR-2) are shown in Table 4 (Initial Conditions). The initial phosphate concentrations strongly affect the theoretical (fitted) and experimental results for the two types of wastewater (MR-1 and MR-2).

**Table 4.** Adsorption results for wastewater using GFH for different initial phosphate concentrations, $C_0$, and GFH doses.

| Sample | Initial Conditions | | Fitted Values [1] | | | Experimental Values [2] | | |
|---|---|---|---|---|---|---|---|---|
| | $C_o$ (mg/L) | m/V (g/L) | $q_e$ (mg/g) | $C_e$ (mg/L) | $\eta$ (%) | $q_e$ (mg/g) | $C_e$ (mg/L) | $\eta$ (%) |
| MR-1 | 3.1 | 2 | 1.38 | 0.34 | 89.1 | >1.28 | <0.5 | >83.9 |
| MR-1 | 3.1 | 3 | 0.96 | 0.23 | 92.5 | >0.87 | <0.5 | >83.9 |
| MR-2 | 41.6 | 2 | 17.3 | 6.98 | 83.2 | 17.8 | 5.55 | 86.6 |
| MR-2 | 41.6 | 3 | 12.5 | 4.20 | 89.9 | 12.8 | 2.85 | 93.1 |

[1] Use of isotherm and mass balance for GFH and synthetic samples (see Figure S5); [2] average of two values.

Despite the final phosphate concentration for MR-1 remains uncertain due to limitations of the chromatographic method, the experimental $q_e$ and $C_e$ values for the two samples are very close to the theoretical values calculated from the GFH isotherm (see Figure S5). Table S2 shows the initial (t = 0 h) and the equilibrium concentration (t = 120 h) of chlorides, nitrates, and sulfates for the same two samples of wastewater. As could be seen in Table S2, after 120 h of contact, the concentrations of chloride and nitrate remained constant and the concentration of sulfate increased with the dosage of GFH, which is linked to a leaching effect from GFH.

As the fit of the model (obtained with aqueous samples) to wastewater concentrations is very good and the main anions did not decrease with contact time, the potential interferences for adsorption isotherms can be neglected, and this adsorbent could be used for a similar kind of wastewater as those used in the study. The low effect of chloride, nitrate, and sulfate interferences has been reported with multicomponent samples in column experiments [1,19] and batch isotherms [48].

### 3.4. Effect of Particle Size on Kinetics

The adsorption kinetics for the three different materials are shown in Table 5 using the data points from Figure S6a,b. When comparing the pseudo first- and pseudo second-order fits (Equations (8) and (9)), the latter yielded a better correlation and less fitting error. Figure 3 shows the fitted kinetics for the three adsorbents using the pseudo second-order linear model (Equation (9) and Table 5). A detailed graphic of the same fitting in the first 6 h is also shown in the same figure.

Again, the fit of Figure 3 (asymptotic values) and Figure S6b (slopes) yielded a reasonable difference in $q_e$ for milled adsorbents versus FGH and OF-U.

Table 6 lists the fitting parameters to perform the comparison of $1/q_e$ with the two statistical methods. The corresponding CIs for the three adsorbents show that the $1/q_e$ values for the OF-M kinetics fit differ from the slopes of the GFH and OF-U fits. The application of the dichotomous model (Table 5 and Table S3) shows a significant difference

in $1/q_e$ when GFH and OF-M were compared and no significant difference between GFH and OF-U.

**Table 5.** Fitting parameters for the pseudo first- and pseudo second-order adsorption kinetics models.

| | | GFH | OF-M | OF-U |
|---|---|---|---|---|
| | | Pseudo first-order (linear) | | |
| $R^2$ | | 0.6637 | 0.8764 | 0.7313 |
| $k_1$ | $(h^{-1})$ | 0.0189 | 0.0237 | 0.0260 |
| Ln $(q_e)$ | (Ln(mg/g)) | 1.796 | 1.302 | 0.796 |
| Initial rate [1] | (mg/g·h) | 0.27 | 0.31 | 0.36 |
| $SS_{err}$ [1] | | 1055 | 931.1 | 1397 |
| | | Pseudo second-order (linear) | | |
| $R^2$ | | 0.9915 | 0.9989 | 0.9997 |
| $q_e$ | (mg/g) | 14.28 | 13.02 | 13.96 |
| $k_2$ | (g/mg·h) | 0.0146 | 0.0322 | 0.0660 |
| Initial rate [1] | (mg/g·h) | 2.98 | 5.45 | 12.86 |
| $SS_{err}$ [1] | | 85.17 | 58.33 | 28.29 |
| | | Pseudo second-order (non-linear) | | |
| $q_e$ | (mg/g) | 13.66 | 12.21 | 13.52 |
| $k_2$ | (g/mg·h) | 0.0257 | 0.0911 | 0.1437 |
| Initial rate [1] | (mg/g·h) | 4.80 | 13.58 | 26.27 |
| $SS_{err}$ [1] | | 53.34 | 11.47 | 5.88 |

[1] Calculated from the direct fitting of data to kinetic model with $q_e$ and $k_1$ or $k_2$.

Again, these results indicate a loss of adsorption capacity of OF-M compared with GFH and OF-U and the difference between the GFH and OF-U results is non-significant. The pattern for $q_e$ is consistent with those for $q_{max}$. As a result, the GFH and OF-U isotherms can be considered similar, while OF-M differs when the equilibrium values are being considered.

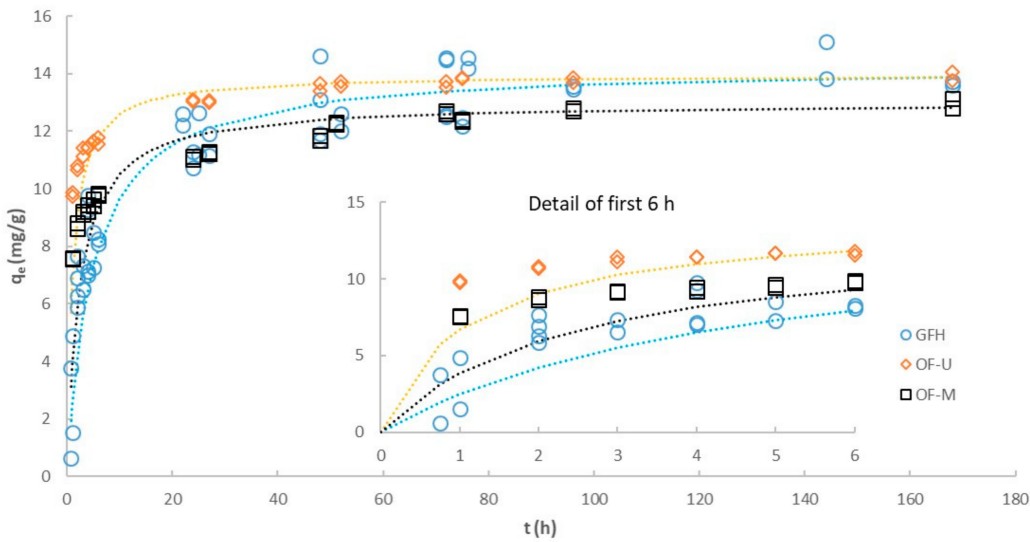

**Figure 3.** Adsorption kinetics for the three tested adsorbents with pseudo second-order fitted linear model (dotted lines, see Table 5).

**Table 6.** 95% confidence interval (CI) for $1/q_e$ and $b_3$ linear fitting parameters in second-order kinetics.

|  |  | **GFH** | **OF-M** | **OF-U** |
|---|---|---|---|---|
|  |  | Linear second-order model CI | | |
| Slope ($1/q_e$) | (g/mg) | 0.070023 | 0.076830 | 0.071627 |
| Standard error | (g/mg) | 0.00979 | 0.000492 | 0.000227 |
| d.f. [1] | - | 44 | 26 | 26 |
| t-Student $_{0.025}$ | - | 2.015 | 2.056 | 2.056 |
| $1/q_e$ UCL | (g/mg) | 0.071996 | 0.077841 | 0.072095 |
| $1/q_e$ LCL | (g/mg) | 0.068049 | 0.075818 | 0.071160 |
|  |  | Dichotomous model comparison with GFH | | |
| $b_3$ UCL [2] | - | - | 0.009413 | 0.004121 |
| $b_3$ LCL [2] | - | - | 0.004200 | −0.000911 |

[1] Degrees of freedom [2]; see Supplementary Materials Table S3.

The detailed kinetic data values of Figure 3 (6 first hours) clearly point to the idea that kinetics is faster for reduced materials than for GFH. The fitting of these initial hours using the second-order linear model of Figure 3 is very poor and underestimates the values of $k_2$ and initial slope. In order to improve this fitting for a better calculation of kinetic parameters, a second-order non-linear model (Equation (7)) with the same data was used. Table 5 compares the values of $q_e$, $k_2$, and initial rate ($k_2\ q_e^2$) for linear and non-linear second-order cases. It could be seen that $SS_{err}$ is reduced when applying the non-linear model; therefore, the non-linear results describe the kinetics much better. Figure 4 shows the new fitting of kinetics to the same data shown in Figure 3, but with non-linear parameters with a visible improvement of the first 6 h.

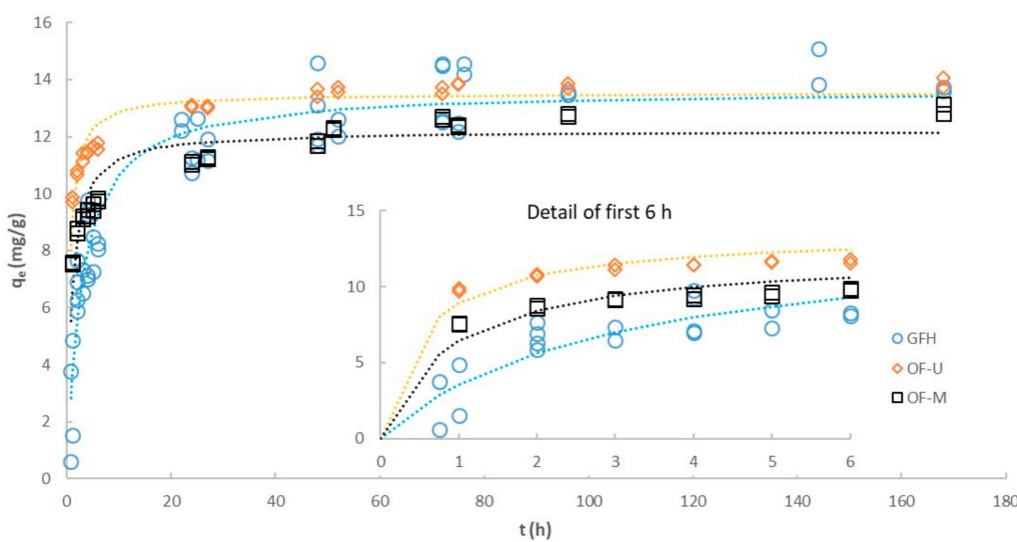

**Figure 4.** Adsorption kinetics for the three tested adsorbents with pseudo second-order fitted non-linear model (dotted lines, see Table 5).

Taking the linear values of $k_2$ and initial rate as a reference, it could be seen that $k_2$ and initial rate are underestimated by a factor of 2–3 times. Despite this difference, the results of Table 5 clearly indicate that $q_e$ values are similar for linear and non-linear models. The calculated value of $q_e$ from GFH Langmuir isotherm fitted values (Table 2 and Figure S5) applied to Equations (1) and (4) was 12.8 mg/g, which is closer to non-linear fitting $q_e$. For all these reasons, fitting values of non-linear kinetic model were chosen for discussion.

For small-sized adsorbents, the adsorption rate in the first few hours is higher compared with granular adsorbents (Figure 4). The values of $k_2$ of OF-M compared with GFH increased 3.5 times and the values of $k_2$ of OF-U compared with GFH increased 5.6 times.

Comparing the initial slope, the values of OF-M compared with GFH increased 2.8 times and the values of OF-U compared with GFH increased 5.5 times. As a final remark, the experimental results indicate that the solid OF-U performed better because the isotherm was similar to GFH and $k_2$ could be increased 5.6 times to reach the equilibrium value ($q_e$) in less time.

This trend linking size reduction and increment of kinetic rate was observed in three different GFH materials [17]. The key point highlighted in this reference was the importance of higher fraction of mesopores bigger than 10 nm for the enhancement of adsorption kinetics.

## 4. Conclusions

The present study investigated the potential of size reduction of a low-cost granular adsorbent using milling and ultra-sonication on the batch-scale adsorption of phosphate. While the milling procedure was more effective for size reduction, the total elimination of the initial GFH structure led to a notable decrease in the specific surface area and $q_{max}$ and $q_e$. Size-reduction using ultra-sonication partially eliminated the initial GFH structure and part of its specific surface area, but $q_{max}$ and $q_e$ remained similar to the GFH. The effect in milled adsorbent could be linked to the high loss of specific surface area (50%) and, specifically, to the 100% loss of micropore area measured. The micropore area has been pointed to as an important contributor for phosphate adsorption in Ferrosorp material [17]. The use of ultra-sonication is linked to moderate size reduction and partial disaggregation, keeping part of the micropore area. Ferrosorp has shown, in other references [17], that the adsorption capacity of FGH is maintained after a moderate grinding process.

For the two size-reduced adsorbents, an important increase of the initial rate and $k_2$ was observed when compared with the GFH. This effect has also been reported for phosphates using Ferrosorp, and it is linked to the limitation of small pores' diffusion (below 10 nm) in the case of granular sorbents [17]. Kinetic mechanisms in granular adsorbents take place along the different transfer zones and include bulk fluid transport, film transport and intra-particle diffusion (or intra-pore), and physical attachment [49]. In the case of phosphates, similar mechanisms occur and film transport and intra-pore diffusion are the rate limiting steps [14]. Size reduction procedures decrease the intra-pore effect and increase the overall kinetics, which is measured as a pseudo-second order law.

Owing to this increase in rate, q values from kinetics will approach $q_e$ and the isotherm will play a relevant parameter in the removal of phosphate. In the case of wastewater, the potential presence of interferences during adsorption also plays an important role. Future investigations open the possibility of developing a new strategy that could be applied to improve low-cost GFH quality adsorbents and enable the re-use of GFH phosphate-loaded adsorbents after its size reduction.

**Supplementary Materials:** The following are available online at https://www.mdpi.com/article/10.3390/w13111558/s1, Figure S1: Size distribution (a) OF-M (milled), (b) OF-U (ultra-sonicated); Figure S2: BET curves (a) GFH, (b) OF-M (milled), (c) OF-U (ultra-sonicated); Figure S3: Regression fitting for the calculation of $pH_{PZC}$ (GFH); Figure S4. Linear regression fittings for the equilibrium models (a) Freundilch model, (b) Langmuir model; Table S1. 95% significance test for $b_3$ in Langmuir fitting parameters (a) comparison of GFH (D = 0) and OF-M (D = 1), (b) comparison of GFH (D = 0) and OF-U (D = 1); Table S2. Adsorption of main anions in wastewater; Figure S5. Theoretical phosphate removal from Langmuir isotherms (GFH) as a function of initial phosphate concentration and adsorbent concentration (dosage); Figure S6. Linear regression fittings for the kinetic models (a) pseudo first-order model, (b) pseudo second-order model; Table S3. 95% significance test for $b_3$ in second-order fitting parameters, (a) comparison of GFH (D = 0) and OF-M (D = 1), (b) comparison of GFH (D = 0) and OF-U (D = 1).

**Author Contributions:** Conceptualization, V.M. and I.J.; methodology, V.M. and J.A.B.; investigation, D.R. and B.F.; resources, V.M. and I.J.; writing—original draft preparation, V.M.; writing—review and editing, V.M. and I.J. All authors have read and agreed to the published version of the manuscript.

**Funding:** This research was funded by the Spanish Ministry of Science, Innovation, and Universities and Agencia Estatal de Investigación/European Regional Development Plan (grant CGL2017-87216-C4-3-R). Researchers from Eurecat were financially supported by the Catalan government through the grant ACCIÓ-Eurecat (Project PRIV2019-MICONANO).

**Institutional Review Board Statement:** Not applicable.

**Informed Consent Statement:** Not applicable.

**Data Availability Statement:** The data presented in this study is available on request from the corresponding author.

**Acknowledgments:** Authors wish to thank HeGo Biotec GmbH for supplying Ferrosorp GW® samples and Neus Bahí from EURECAT for the analysis of the samples. Editorial assistance, in the form of language editing and correction, was provided by XpertScientific Editing and Consulting Services.

**Conflicts of Interest:** The authors declare no conflict of interest. The funders had no role in the design of the study; in the collection, analyses, or interpretation of data; in the writing of the manuscript; or in the decision to publish the results.

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
