# Peer review of "Improvement of Phosphate Adsorption Kinetics onto Ferric Hydroxide by Size Reduction"

_water, doi:10.3390/w13111558_

Round 1

Reviewer 1 Report

The manuscript should be accepted after the following corrections have been implemented.

  • Lines 167, 179 - citations are not correct. The works in which Froundlich and Langmuir models (line 167) or kinetic models (line 179) were used are presented as a reference, and not the authors of these models (correct references for equation 4 Langmuir I. (1916): The constitution and fundamental properties of solids and liquids, Journal of the American Chemical Society 38, 2221 – 2295)
  • In my opinion chapter 3.4 Effect of Particle Size of Kinetics should be placed before chapter 3.3. Removal of phosphate from wastewater
  • Table 5 should also include the data for the pseudo-first order model obtained from Equation 8 (PFO linear form)
  • Lines 153 – 164 - The equilibrium experiment is too broadly described.

Author Response

Manuscript Water-1236094: responses to referee #1

Dear reviewer #1:

In the new version of manuscript Improvement of Phosphate Adsorption Kinetics onto Ferric Hydroxide by Size Reduction you could find in “track changes” the new aspects you asked for revision.

  • Classical references on Langmuir and Freundlich models from the original authors have been added. Changes in the list highlighted in yellow.
  • We have included the additional data of pseudo first-order kinetic (calculated values of initial rate and SSerr given by correlation). The fittings of Table 5 take the data points from Supplementary Materials. The figures S6a and b show the linear fitting.
  • The description of equilibrium in lines 153-164 was mainly due to the origin of sample description. This part has been advanced after the initial synthetic samples explanation, and we have re-written the procedure of wastewater part, removing the conditions of room temperature and stirring, that were mentioned before.
  • The reason because we placed description and results of wastewater and GFH in equilibrium part is because we performed only equilibrium experiments with GFH (as we explained in previous cover letter dated 10th May). We have the results in two blocks: equilibrium (synthetic with 3 adsorbents and wastewater with GFH) and kinetics (synthetic samples with 3 adsorbents). I think that, as we calculated the removal from equilibrium, is the best place and, in this way, we keep the kinetic part as an important block for discussion. That is because I have maintained this part in the same place in the new version, because for me is difficult to replace it after the kinetics, because maybe it will be expected kinetic experiments with wastewaters and the 3 adsorbents, that is not the case.

Do not hesitate to contact again if new changes are requires

Best regards

Reviewer 2 Report

The manuscript was ameliorated. Thank you for your effort. I recommend it for publishing. An advice don't ommit the physical chemical analysis of your samplew in the future.

Author Response

Manuscript Water-1236094: 

Dear reviewer #2:

Thank you for your comments. We will not forget to include a better characterization of the real samples.

Best regards

Round 2

Reviewer 1 Report

Thank you very much,
currently, well done at the moment

This manuscript is a resubmission of an earlier submission. The following is a list of the peer review reports and author responses from that submission.

Round 1

Reviewer 2 Report

This manuscript by V. Marti et al. aims to assess the phosphate adsorption onto ferric hydroxide.

The novelty is not the high point of this study as hundreds of paper already investigated the adsorption of phosphate. However, it would be suitable for Water in case of correct experiments and discussions. Yet, it is not the case here.

The introduction does not lead to the purpose of the work.

The purpose and scope of the work are not clearly defined.

Many references are cited poorly. For example, references [21, 22] are completely unrelated to the topic of the work.

No reference is made for kinetic and isothermal models.

The kinetic models that were used to evaluate the sorption kinetics are incorrectly named.

The research methodology is presented very chaotically, for example, the terms used by the authors for the tested materials are changed.

The kinetic and equilibrium experiments are conducted under completely different conditions.

The constants of isothermal models and kinetic models are calculated on the basis of linear equations and presented in the figures in general forms.

Research results are incomplete, sometimes presented in the text of the article, sometimes as a supplement. Some are completely absent.

From all these mistakes, it is very hard to clearly assess the potential of this material for phosphate removal.

As a consequence, I recommend to reject this manuscript which is below the publication standard of Water.